# KRAS Pathway Alterations in Malignant Pleural Mesothelioma: An Underestimated Player

**DOI:** 10.3390/cancers14174303

**Published:** 2022-09-02

**Authors:** Lilith Trassl, Georgios T. Stathopoulos

**Affiliations:** 1Institute for Lung Health and Immunity, Helmholtz Munich-German Research Center for Environmental Health, 81377 Munich, Germany; 2Munich Medical Research School, Ludwig-Maximilian-University Munich, 81377 Munich, Germany; 3German Center for Lung Research, 35392 Giessen, Germany

**Keywords:** mutations, receptor tyrosine kinase pathway, TP53, RAS, PI3K, MAPK

## Abstract

**Simple Summary:**

Malignant pleural mesothelioma (MPM) is a rare, incurable cancer. *KRAS* pathway alterations are frequent in human MPM but have been likely underestimated by next generation sequencing studies.

**Abstract:**

Malignant pleural mesothelioma (MPM) is a rare, incurable cancer of the mesothelial cells lining the lungs and the chest wall that is mainly caused by asbestos inhalation. The molecular mechanisms of mesothelial carcinogenesis are still unclear despite comprehensive studies of the mutational landscape of MPM, and the most frequently mutated genes *BAP1, NF2, CDKN2A, TP53,* and *TSC1* cannot cause MPM in mice in a standalone fashion. Although *KRAS* pathway alterations were sporadically detected in older studies employing targeted sequencing, they have been largely undetected by next generation sequencing. We recently identified *KRAS* mutations and copy number alterations in a significant proportion of MPM patients. Here, we review and analyze multiple human datasets and the published literature to show that, in addition to *KRAS*, multiple other genes of the *KRAS* pathway are perturbed in a significant proportion of patients with MPM.

## 1. Introduction

Malignant pleural mesothelioma (MPM) is a rare cancer of the mesothelium lining the pleural space that rapidly progresses and is still uncurable [1,2]. It is associated with asbestos exposure and develops after a long latency period of approximately 40 years [1]. Despite the exact mechanism of pathogenesis and carcinogenesis still being unclear, it is believed that upon inhalation of long and thin asbestos fibers, these penetrate the pleural space and interact with mesothelial and immune cells, causing repeated inflammation and tissue damage and repair, and thus carcinogenesis by several possible mechanisms [1,2]. These include reactive oxygen species (ROS) that cause DNA (deoxyribonucleic acid) damage, aneuploidy and abnormal chromosomal structure caused by the physical disruption of mitosis, carcinogenic molecules bound to the fibers, and the release of cytokines and growth factors upon fiber interaction [2]. Although the use of asbestos has been restricted in more developed countries, its historical presence in older homes and buildings still poses a source of exposure during maintenance, reconstruction works/renovations, or abatements today. In addition, the long latency period and continued use of asbestos worldwide will ensure that MPM continues to be a major health concern [1,3,4]. The exact global magnitude of the asbestos pandemic is difficult to determine, but an annual average of approximately 14,200 mesothelioma cases worldwide has been estimated from 2008 data [3]. Due to the late onset of symptoms, MPM is often diagnosed at a late stage, with a median survival of 9 to 18 months [5,6]. The symptoms include chest pain and breathlessness, which are often caused by pleural effusions, i.e., exudative fluid accumulations in the pleural cavities, limiting respiratory movement [1]. The diagnosis usually includes radiological imaging and biopsies, which are then also used for histological classification to one of the three MPM subtypes, i.e., epithelioid, sarcomatoid, or biphasic [1,7]. Treatment options include chemotherapy, radiotherapy, and surgery, or a combination of these in a tri-modal approach. The standard chemotherapy treatment is a combination of cisplatin and pemetrexed, but response rates are limited [1]. Recent advances in therapy options, which significantly prolong the overall survival of patients, shed hope. These include the Checkmate 743 regimen ivolumab plus ipilimumab for patients with unresectable MPM [8], hyperthermic intrathoracic chemoperfusion of thoracic cavity (HITHOC) after pleurectomy and decortication (P/D) for patients with localized MPM [9], or the use of bevacizumab in addition to standard chemotherapy [6]. Nevertheless, there is a need for the development of more therapeutic options that will further increase the survival of MPM patients. 

A key to better therapies is a deeper understanding of the mutagenic processes of MPM, which is complex and challenging. Although multiple efforts have been directed towards unravelling the molecular landscape, contributing pathways, initiators, and evolution of this cancer type, the mechanisms of MPM evolution are still largely unclear. The molecular landscape of MPM is characterized by high inter-patient and intra-tumor heterogeneity, comparably low somatic mutational burden, frequent loss-of-function of tumor suppressors (*BAP1, NF2, CDKN2A, TP53, TSC1,* etc.), and occasional gain-of-function of proto-oncogenes (*PIK3CA, EGFR, KRAS, NRAS, HRAS, BRAF*, etc.) [2,4,10,11,12,13,14,15,16,17]. Interestingly, single tumor suppressor deletion in the pleural or peritoneal mesothelium of mice does not suffice to trigger MPM [2,4,10,11,12,13,14,15,16,17]. On the other hand, we recently discovered a cardinal role for KRAS signaling in MPM. In this article we identified low allelic frequency *KRAS* mutations by sensitive digital droplet polymerase chain reaction (PCR) in 25% of our MPM patients from Germany, copy number alterations in up to half of primary MPM cell lines from France, and found that ectopic *KRAS*^G12D^ overexpression with or without *Trp53* deletion in the pleural and peritoneal mesothelium of mice led to disease identical to human MPM [18]. 

The Ras (rat sarcoma) signaling pathway plays an important role in mammalian cell proliferation, and across human cancers, mutations of the Ras family (*NRAS*, *HRAS*, and *KRAS*) are the most widespread [19,20]. Ras proteins are small GTPases that can switch between an active guanosine triphosphate (GTP)-bound state and an inactive guanosine diphosphate (GDP)-bound state. They are activated by receptor tyrosine kinases such as the epidermal growth factor receptor (EGFR) family. In their active form, Ras proteins activate several downstream signaling cascades relevant for gene regulation, proliferation, and apoptosis evasion, including the mitogen-activated protein kinase (MAPK) and phosphatidylinositol 3-kinase (PI3K) pathways. *RAS* mutations frequently lead to a disruption of the molecular switch, causing it to be consistently active and stimulating the downstream pathways. Furthermore, without *RAS* mutations themselves, faulty Ras pathway activation can be achieved by mutations or alterations in the other pathway agents, up- and downstream of Ras [19,20]. For example, it has been shown that NF2 (neurofibromatosis 2 or Merlin) serves as a tumor suppressor by inhibiting KRAS and that NF2 mutations can cause persistent Ras signaling [21]. Furthermore, mutant *KRAS* has been demonstrated to activate the apoptotic MST2-LATS1 serine/threonine-protein kinase 3/STK3—large tumor suppressor kinase 1 pathway by binding the tumor suppressor RASSF1A (Ras association domain-containing protein 1). In this pathway, LATS1 then induces apoptosis by binding MDM2 (mouse double minute 2 homolog), therefore stabilizing p53 (tumor protein p53). However, this is antagonized by autocrine EGFR activation and wild-type KRAS stimulation of the AKT (protein kinase B) pathway, which inhibits the MST2 pathway [22]. Furthermore, functional relationships of p53 with BAP1 (BRCA1 associated protein-1) and one of the *CDNK2A* proteins p14^ARF^ have been reported, showing that p53 controls BRCA1 (breast cancer type 1 susceptibility protein) and p14^ARF^ expression, and is stabilized by p14^ARF^ via MDM2 binding [23,24,25]. In addition, Ras pathway activation has been frequently observed in MPM [19], and in the past, we could report a *KRAS^G12C^* mutation in an asbestos-induced MPM cell line [26]. 

Based on the above findings and a functional report of the interconnectedness of the KRAS and TP53 pathways [22], we hypothesize that KRAS pathway alterations might represent a widely underestimated player in MPM. In addition to our original research report, analyzing *KRAS* and *TP53* mutations in human and experimental MPM [18], here we examine the whole KRAS pathway as proposed elsewhere [22] (indicated in Figure 1a and including the following genes: *KRAS, EGFR, NF2, BRAF, MAP2K1, MAPK1, PIK3CA, AKT1, RASSF1, STK3, LATS1, MDM2, TP53*). For this, the relevant literature is reviewed, and publicly available datasets are analyzed as follows.

## 2. Materials and Methods

The literature reviewed in this article was either already known and cited by our group or obtained by searching PubMed with multiple combinations of the keywords stated below and setting the display options to best match. The keywords used were mesothelioma, malignant pleural mesothelioma, genome, transcriptome, sequencing, KRAS, Ras, review, molecular classification, cell lines, characterization, clonal evolution, global incidence. The publications for this article were chosen by firstly reading the title and the abstract and evaluating if the content fitted the query. Next, the methods section was analyzed, and the publication categorized to the topics of this article. Then, the studies were read and the most relevant were chosen to be reviewed here. The methods and datasets used for cBioportal and microarray analyses are given in the relevant figures. The integrative genomics viewer [27] was used to detect the copy number alterations presented in Figure 1a.

## 3. Review Body

### 3.1. KRAS Pathway Alterations in Older Studies of MPM

Due to the high relevance and frequency of *KRAS* mutations in other cancers, especially lung cancers, early studies sought to detect these in MPM, utilizing targeted approaches. Metcalf et al. reported in 1992 that they were not able to detect any *KRAS* mutations by Sanger sequencing (codons 12, 13, and 61) and therefore concluded that *KRAS* does not play an important role in MPM development. With northern blot and immunocytochemistry, the authors also detected *TP53* mutations in two individuals with high protein expression, whereas expression loss was detected in one individual with wild type p53. As the *TP53* alterations were seen as rare and neither the mutation nor expression status was correlated with the tumorigenicity of these cell lines, the authors came to the same conclusion as for *KRAS* [28]. In 2000, Ni et al. also did not detect any *KRAS* mutations by PCR-primer-introduced-restriction-site assay (PCR-PIRS) and Sanger sequencing validation (codons 12, 13, and 61) of asbestos-exposed human and rat mesothelioma tissues. Furthermore, no mutations of *TP53* (exons 5–8) were found, and so the authors came to the conclusion that *KRAS* and *TP53* genes do not play a critical role in MPM development [29]. On the other hand, Patel et al. were able to report Ras pathway activation in human mesothelioma cell lines in 2007. Although they did not find *KRAS* mutations, they were able to show higher levels of Ras-GTP and known Ras effectors such as ERK1/2 (extracellular signal-regulated kinases). Additionally, p38 MAPK (mitogen-activated protein kinases) were found to have a higher activity in comparison to a non-transformed mesothelial cell line. The inhibition of JNK (c-Jun N-terminal kinases) or ERK decreased cell proliferation, although it did not induce apoptosis [19]. In 2013, Mezzapelle et al. set out to analyze the frequency of *EGFR* mutations and some of its downstream effectors (*KRAS*, *BRAF* and *PIK3CA*), based on previous reports indicating an overexpression of EGFR in 60–70% of MPM tissue specimens. They were able to report *KRAS* (5/77), *BRAF* (3/77), and *PIK3CA* mutations (1/77), but no cases of *EGFR* mutation. The authors saw no difference in patient survival, but concluded that the mutation of downstream effectors of EGFR was not rare, with an incidence of 12% in their study [16]. In a broader approach, Bott et al. analyzed copy number alterations to identify altered genes, which were then subjected to Sanger sequencing. The authors did not analyze *KRAS*, but found frequent homozygous deletion of *CDKN2A* and heterozygous deletion and mutation of *NF2*, in line with other studies of the time, reporting these two as the most frequently altered tumor suppressor genes [2,13]. Apart from these, the authors also detected frequent *BAP1* (12/53) and *LATS1* (2/53), *LATS2* (2/53), *RASSF1* (1/53), and *TP53* (3/53) mutations, among others [13]. Taken together, evidence of Ras pathway alterations arise from early studies of MPM genomes.

### 3.2. Molecular Alterations in Published Next Generation Sequencing Studies of MPM

With the new technology of next generation sequencing (NGS) many studies trying to unravel the molecular landscape of MPM were conducted, utilizing a combination of whole genome, whole exome, ribonucleic acid (RNA), and methylome sequencing techniques. One of the first to publish a whole-exome analysis of MPM by NGS were Guo et al. in 2014. The most frequently mutated or altered genes they found were *BAP1* (8/22), *NF2*(3/22), and *CDKN2A* (10/22). *TP53* mutations were also detected in two cases. The most significantly altered pathways determined in this study were the cell cycle (12/22) and MAPK (11/22) pathways [12]. Kato et al. sequenced different malignant mesotheliomas from 42 patients (23 pleural) by NGS. These authors also identified *BAP1*, *NF2*, and *CDNK2A* to be the most frequently altered genes, and did not detect *KRAS* mutations [14]. In another study with specimens from 10 MPM patients, the findings from NGS were correlated to gender and histology subtype, showing lower *BAP1* mutation rates in sarcomatoid MPM and more frequent *TP53* mutations in women. The pathway that was the most significantly affected was the integrin-linked kinase (ILK) pathway, as five samples showed mutations of at least one of its agents (*MYH9, MYH6, MYH10, PIK3C2A, RHOA,* and *TNFRSF1A*). *KRAS* mutations were not detected [17]. Lo Iacono et al. used a targeted NGS approach and found the most frequent mutations in genes of the p53/DNA repair pathway (including *TP53*, *SMACB1*, and *BAP1*), and the PI3K pathway (*PDGFRA, KIT, KDR, HRAS, PIK3CA, STK11,* and *NF2).* Specific *TP53* variants were correlated to time to progressive disease and overall survival, as was the accumulation of variations. *KRAS, HRAS*, and *NRAS* mutations were detected in 14, 75, and 5 patients (of 123) respectively, with an allelic frequency of >10% [15]. In a comprehensive study by Bueno et al. the transcriptome, whole exome, and targeted exomes from 211, 99 and 103 MPM tumors were analyzed, respectively. Among the most significantly mutated genes that were found were *BAP1, NF2, TP53, SETD2,* and *SETDB1*. Although no *KRAS* and *HRAS* mutations were identified, recurrent mutations (hotspots) for *NRAS* and *TP53* were determined. Furthermore, patients with *TP53* mutations presented an overall lower survival than those with wild-type (WT) *TP53* [11]. As part of The Cancer Genome Atlas (TCGA), Hmeljak et al. composed a comprehensive integrated genomic study of 74 MPM samples. The genes they found to be significantly mutated were *BAP1, NF2, TP53, SETD2,* and *LATS2. CDKN2A* loss was strongly associated with shorter overall survival, whereas *BAP1* mutation was not. When the authors analyzed the phenomenon of genome-wide loss of heterozygosity in their cohort, they identified three cases where more than 80% of the genome was affected by this. These patients, and another two of an additional cohort, were grouped to a novel subset presenting genomic near-haploidization, (except for chromosomes 5 and 7, which remained heterozygous), mainly female gender (4/5), *TP53* mutations (4/5), and no histologic subtype association. Genomic near-haploidization is usually followed by genome doubling events; therefore, it could be determined that in these patients, most mutations including *TP53* occurred before the genome duplication. With a multiplatform molecular profiling approach, the authors identified four molecular subtypes of MPM significantly associated with patient survival. *TP53* and *NF2* mutations and *CDKN2A* homozygous deletions were found among all clusters, but were not equally distributed [4]. Collectively, the above studies limited enthusiasm on the presence of KRAS pathway mutations other than *NF2* or *TP53* in mesothelioma.

### 3.3. Transcriptomic MPM Studies

For a better understanding of the mechanisms of MPM development and progression, several transcriptomic studies focusing on the gene expression patterns and contributing pathways were conducted. In 2014 Suraokar et al. performed gene expression analysis of 53 human MPM tumors by microarray. The authors identified the metaphase checkpoint pathway, including the mitotic spindle assembly checkpoint (MSAC) pathway and kinetochore genes, to be the most significantly altered. Of these, 18 components including *MAD2L1* and *AURKA* were upregulated in their study. Network analysis showed the cell cycle and microtubules network to be the most affected. Three molecular clusters of differentially expressed genes were reported, which were not associated to histologic subtype and overall survival. RNA sequencing (RNAseq) analysis from eight tumors of the subgroup with the highest expression of MSAC genes, did not identify any mutations in the genes of this pathway. Small molecule inhibitors of some of the pathway components were not effective in MPM cell lines [30]. In a study already mentioned above, Bueno et al. also performed transcriptomics as part of their comprehensive analysis. From RNA sequencing data, the authors obtained four molecular subtypes, which they referred to as sarcomatoid, epithelioid, biphasic- epithelioid (biphasic-E), and biphasic-sarcomatoid (biphasic-S). The sarcomatoid cluster included all histologically sarcomatoid samples, but also epithelioid and biphasic. The molecular epithelioid cluster mainly consisted of histologically epithelioid samples and one biphasic, but only contained 38% of all histologically epithelioid samples. The rest were distributed among the other clusters, but also showed a significantly lower overall survival than the other histologically epithelioid samples categorized to the molecular epithelioid cluster. The epithelioid and sarcomatoid clusters had the most distant gene expression patterns with about 200 up- and downregulated differentially expressed genes. The authors also identified several gene fusions (including among others *NF2, BAP1, SETD2, STK11*) and splice variants by RNAseq analysis. Several significantly altered pathways, including Hippo, mTOR (mammalian target of rapamycin), and p53 signaling were determined by integrated analysis including mutation, gene expression, copy number alterations, and fusion data [11]. Hmeljak et al. used a multi-omics integrated approach to identify four subtypes in the TCGA dataset (mentioned above) associated with patient survival. The cluster with the worst prognosis was defined by enrichment of *LATS2* mutations and homozygous *CDKN2A* deletions, higher leukocyte fraction, Th2 immune cell signature, higher mRNA (messenger RNA) expression of Aurora kinas A (AURKA), E2F transcription factor targets, G2–M checkpoints, DNA damage response genes, and upregulation of the PI3K–mTOR and Ras–MAPK signaling pathways [4]. 

In 2019 Blum et al. analyzed 63 MPM tumor samples by microarray and found two molecular subtypes by unsupervised hierarchical clustering that were associated with prognosis and histology. Intra-subtype heterogeneity revealed two groups in each of the subtypes (C1A and B, C2A and B). Comparison with the subtypes identified by others revealed two distinct consensus clusters in all studies, namely an extreme “epithelioid” and “sarcomatoid” subtype, as already described by Bueno et al. All other identified subtypes were portrayed as a mix of both extremes with different gradients. Therefore, the authors then used a novel deconvolution method (weighted in silico pathology, WISP) examining a given tumor sample as a mixture of epithelioid-like, sarcomatoid-like, and non-tumor components (E-comp, S-comp) and estimating their proportions (E-score, S-score). The overexpression of 110 previously reported genes for the histologic subtypes were in accordance with the scores. New genes associated with the E- and S-score were also detected, such as *PDZK1IP1* (a *MAP17* cargo protein) and *AXL* (a tyrosine kinase), respectively. The associated signaling pathways were cell junctions and complement and metabolic pathways for the E-comp, whereas epithelial-to-mesenchymal transition (EMT), p53 signaling, cell cycle, angiogenesis, and immune checkpoints were determined for the S-comp. Interestingly, *TP53* and *NF2* genetic alterations were positively associated with the S-score. Furthermore, a higher S-score was related to poor prognosis, even when it was restricted to the epithelioid histology, and a cut-off value of 22% was able to distinguish patients’ overall survival [31]. Taken together, these studies supply a manifold of information and emphasize the complexity of this malignancy. Strikingly, many of the identified pathways are closely connected to and either influence or are influenced by the KRAS/TP53 pathway. More and in-depth research will be needed to unravel the mechanisms contributing to MPM.

### 3.4. Molecular Alterations in MPM Cell Lines

To classify and determine molecular subtypes of MPM, de Reyniès et al. profiled 38 primary human MPM cell cultures by transcriptomic microarray in 2014 and identified two subtypes (C1 and C2) by consensus clustering. Measurements of 40 genes by qRT-PCR (quantitative reverse transcription PCR) were used to identify a predictor based on the three genes *PPL*, *UPK3B*, and *TFPI.* The findings were validated in 29 additional MPM cell cultures and a cohort of 108 frozen MPM tumor samples. By targeted Sanger sequencing, the mutational profile for *BAP1, CDKN2A, CDKN2B, NF2,* and *TP53* were also determined. The identified subtypes significantly differed in prognosis and were partly related to histologic subtype. All cultures of sarcomatoid histology were clustered to subtype C2, whereas cultures determined as epithelioid were found in both subtypes, though the ones of C2 had a worse prognosis. The epithelial-to-mesenchymal transition (EMT) and TGF-ß (transforming growth factor beta) pathways were reported to be differentially regulated between the two clusters, and *BAP1* mutations occurred more frequently in the C1 cluster [32]. These finding were shown to be consistent with the later conducted classification study by Blum et al., mentioned above [31]. Sneddon et al. performed genomic and transcriptomic characterization of 27 low-passage cell cultures derived from the pleural effusions of MPM patients. Alterations of *BAP1* (70 %), *CDKN2A* (96 %), and *NF2* (67 %) were detected at a higher frequency than reported for tumor biopsies. Interestingly, homozygous loss off all or part of the *CDKN2A* locus was reported in 26 of 27 samples. Furthermore, *LATS2* (59 %) and *TP53* (22 %) alterations were detected at high frequencies. Two samples were reported with homozygous loss of all or part of *TP53*, resulting in no or low expression. Low *TP53* expression was found to be significantly associated with higher overall survival when adjusted for age, sex, treatment, and histology status. Significant losses also occurred in the chromosomal regions 19p13.3, 1p36.32, and 8p23.1, including the genes that encode mTOR and the beta-defensin gene cluster (*DEFB*), associated with small cell lung cancer. The authors highlight the advantages of the pleural effusion derived cell cultures they studied, demonstrating similar molecular characteristics with tumor biopsy samples commonly used and emphasizing the minimal-invasive sampling that can be performed sequentially over time [33]. In a study by Quispel-Jansen et al., 889 different cancer cell lines (both immortalized and primary) were subjected to molecular characterization and extensive drug screening. The authors detected a subgroup of MPM cell lines that were particularly sensitive to fibroblast growth factor receptor (FGFR) inhibition, although no mutations of FGFR family members were detected. Instead, these cell lines exhibited BAP1 protein loss. An association between *BAP1* gene expression loss and increased *FGFR1/3* and *FGF9/18* expression was validated by murine xenograft models and *BAP1* knockdown and overexpression in cell line models [34]. In 2017, we reported the murine malignant pleural mesothelioma cell line AE17 to harbor *Kras*^G12C^ mutations, but no *Egfr, Pik3ca*, and *Braf* mutations. Intrapleural injection of this cell line was able to induce pleural effusions, whereas *Kras* silencing by shRNA (small hairpin RNA) abrogated this. Microarray-based transcriptome analysis then allowed the identification of 25 genes overexpressed in the analyzed *KRAS*-mutant human and mice cell lines in comparison to those cell lines without *KRAS* mutation. Among the transcripts with the highest expression in AE17 were *CCL2* (chemokine C–C motif ligand 2) and *HIST1H1B* (histone cluster 1, H1b) [26]. Overall, similar observations were made in studies of cell lines compared as those studying the tumor biopsies directly, which is promising for future research in the field. 

### 3.5. Findings from Newer Sensitive Methods

In a recently published study by our group, we employed digital droplet PCR (ddPCR) to detect *KRAS* mutations in human MPM in comparison with lung adenocarcinoma samples with a high sensitivity (down to 1:20,000 mutant copies) [18]. In a significant proportion of human MPM samples (25%, 3/12), we were able to detect *KRAS* mutations, despite low copy numbers. Copy number alterations in the form of gains or losses of the *KRAS* locus were also identified in 10 and 2 of 32 primary MPM cell lines from France, respectively. Furthermore, we combined the data from nine of the most significant human genomic studies (755 patients, 1616 mutations) and identified the top 25 mutated genes, including *NF2, BAP1, CDKN2A, TP53, LATS*, etc. Strikingly, the mutation rates of the RAS pathway components (*KRAS, EGFR, PIK3CA*) and p53 pathway components (*TP53, STK11*) were each attributed 10% of all mutations. Additionally, we were also able to report that targeting oncogenic *KRAS*^G12D^ to the murine pleura causes MPM, and in combination with *TP53* loss elicits aggressive MPM with pleural effusions, secondary *BAP1* mutations, and a transcriptome that resembles that of human MPM. Furthermore, these murine MPM were not only transplantable, but also actionable by KRAS inhibition using the phosphodiesterase δ blocker deltarasin, which inhibits membrane binding and hence activation of KRAS [18]. This study shows that, indeed, molecular alterations of *KRAS* and its downstream effectors might be more frequent and important than we currently think.

### 3.6. Occult KRAS Pathway Alterations in Published MPM Datasets

Based on the collective knowledgebase described above, as well as biologic evidence describing a signaling pathway interconnecting *KRAS* and its downstream effectors with *TP53*, we queried all 13 *KRAS/TP53* pathway genes (including *EGFR*, *KRAS*, *NF2*, *BRAF*, *PIK3CA*, *RASSF1*, *MAP2K1*, *AKT1*, *STK3*, *MAPK1*, *LATS1*, *MDM2*, *TP53*) in The Cancer Genome Atlas (TCGA) pan-cancer MPM dataset [4,35] (available at https://www.cbioportal.org/ (accessed on 3 March 2022) using permanent link https://bit.ly/3BypsnC (accessed on 3 March 2022), with *n* = 82 patients). Multiple KRAS pathway mutations and copy number alterations (CNA), as well as three fusions, were observed in 44 affected patients (54% alteration frequency) (Figure 1a). This finding indicated that, although KRAS mutations might be infrequent in human MPM per se, KRAS pathway alterations interrogated integrally are indeed very frequent, affecting half of the patients. This was also evident when *KRAS* pathway mutation frequencies were interrogated in a larger MPM dataset (*n* = 775 patients) from the Catalogue of Somatic Mutations in c=Cancer (COSMIC; dataset available at https://cancer.sanger.ac.uk/cosmic/browse/tissue?wgs=off&sn=pleura&ss=all&hn=mesothelioma&sh=&in=t&src=tissue&all_data=n (accessed on 3 March 2022)), since this allowed analyses stratified by histologic subtype [36]. In COSMIC, the cumulative mutation frequency of the KRAS pathway increased gradually in biphasic and sarcomatoid MPM compared with epithelioid MPM (Figure 1b), suggesting a link between KRAS pathway changes and more aggressive histology. This is in line with our findings from experimental induction of KRAS mutations with or without *Trp53* deletions in the murine pleural and peritoneal mesothelium, which led to biphasic/sarcomatous MPM [18].

We subsequently reanalyzed published microarray data, mentioned before [30] (GEO dataset GSE51024; Suraokar and Wistuba, 2013) in order to determine whether there is transcriptional upregulation of the *KRAS* pathway in MPM. Principal component analysis (PCA) and unsupervised hierarchical clustering of the expression profiles of 55 MPM and 41 normal lung tissues revealed 2204 gene probes that were biologically and statistically significantly differentially regulated, as well as 14 WikiPathways [37] that are perturbed in MPM (Figure 2a–c). Interestingly, the two most significantly altered pathways were “phosphoinositide 3-kinase (PI3K)-protein kinase (AKT) signaling” and “focal adhesion-PI3K-AKT-mTOR signaling” (hereafter collectively called PI3K-AKT signature). Moreover, this PI3K-AKT signature included several transcripts of known importance in MPM biology such as *SPP1* and *FGF9* [34,38,39] that were heavily overrepresented in all three histologic MPM subtypes and were capable of accurately discriminating malignant from benign samples on unsupervised hierarchical clustering (Figure 2d–f). Taken together, these data indicate that *KRAS* signaling is at play in a subset of human MPM.

### 3.7. Studies on the Clonal Evolution of Mesothelioma

Our recent results, taken together with relevant studies reviewed above, raise the question of whether KRAS pathway mutations are early tumor-initiating or late subclonal events during MPM evolution. Only one elegant study focusing on the clonal evolution of mesothelioma by Zhang et al. addressed this issue in 2021 using multi-region sampling [10]. The authors performed whole-exome sequencing of 90 MPM tumor samples derived from 22 patients. The samples were obtained from 4–5 tumor regions of each patient in a standardized manner. The authors observed great inter-patient and intra-tumor heterogeneity, with phylogenetic trees ranging from linear (64%) to highly branched, correlating with the number of mutations and copy number alterations. Putative driver genes undergoing clonal positive selection during early evolution included *NF2, BAP1, SETD2, FBXW7,* and *PRELID1*. Common evolutionary trajectories were found and grouped into five clusters with prognostic significance and increasing complexity. Cluster 5 presented the highest number of early clonal alterations and significantly shorter overall survival and was consistently of the epithelioid subtype. In one patient, a *TP53* mutation was found, and although this was not identified as a driver gene, it was determined to be a clonal event. This patient also had an early *BAP1* mutation, many copy number alterations, and was classified as belonging to Cluster 5. *BAP1* and *FBXW7* mutation or loss of chromosome 3p21 and 4 were determined to always be early clonal events, whereas *NF2* mutation or chromosome 22q loss were late clonal events leading to Hippo pathway inactivation. Although the authors report no *KRAS* and only one missense *TP53* mutation in their patients, careful curation of their supplementary information revealed 12 *NF2* mutations in 11 patients (five of which were protein-altering), one missense *STK3* mutation, and five additional intronic *TP53* deletions of unknown significance, all potentially affecting the KRAS/p53 pathway as proposed by Matallanas et al. [22]. In addition, their data show significant gains and losses in the chromosomal positions of *KRAS* at 12p12.1 (chr12:25,357,180-25,404,863) and *TP53* at 17p13.1 (chr17:7,570,720-7,591,868) in five and eight patients, respectively [10]. Finally, the average read depth of 276 in this study would likely result in underrepresentation of *KRAS* mutations. Hopefully, such elegant and much needed approaches to unravel the clonal evolution of MPM using phylogenetic analyses of multi-region samples can be conducted with enhanced read depths in the future.

## 4. Discussion

Overall, in the studies reviewed in this work, mutations of *KRAS* and its pathway agents were detected more frequently when a targeted approach was used. In the large cohorts using massive parallel sequencing and comprehensive integrated genomics that yield a great amount of data, *KRAS* mutations were detected rarely and never reached significance. In total, the mutations most frequently reported were *BAP1*, *NF2*, and *CDKN2A*. In most publications, the PI3K, mTOR, MAPK, Hippo, and p53 pathways were determined as the most significantly altered. Most of these are either directly activated by KRAS, are integral parts of the KRAS pathway, or indirectly affect it. The most frequently observed mutated genes belonging to these pathways include *PIK3CA, NF2, LATS1, LATS2, TP53,* and *BAP1*. *RASSF1* and *BRAF* mutations were also more commonly detected in targeted approaches. Various attempts for MPM classification and pathway analysis based on transcriptomics (or integrated analysis) were composed. These revealed multiple differentially expressed pathways including, in addition to the aforementioned, cell junction, epithelial-to-mesenchymal transition, and metaphase checkpoint pathways, among others. Several different clusterings were proposed, some of which were combined by Blum et al. into one model of two extreme components, classifying MPM heterogeneity as a gradual continuum of both [31]. In the reviewed cell line studies, overall similar results concerning the genomic and transcriptomic patterns were observed, with higher mutation frequencies in pleural effusion-derived cells. From our own analysis presented in this review, based on the literature and TCGA, COSMIC, and mentioned GEO datasets, we depicted a KRAS pathway, its alterations, and their mutation frequencies per histologic subtype, and identified the PI3K/AKT pathway as significantly altered and capable of discriminating MPM from normal lung tissue. Furthermore, in a recently published study, we identified *KRAS* mutations and copy number alterations in a significant proportion of human MPM and showed that *KRAS* and *TP53* gene alterations each contribute 10% of all mutations detected in nine combined published studies [18]. 

In another published TCGA pan-cancer pathway analysis, the authors reported 9% and 21% alteration frequencies for the receptor tyrosine kinase RTK/Ras and p53 pathways, respectively, in MPM according to their genomic pathway mapping approach [40]. Very recently, Singh et al. reported the therapeutic potential of a microRNA (short, interfering, non-coding RNA), namely miR-206, that targets *KRAS* as well as agents of the RTK-Ras-MAPK-PI3K/Akt-CDK pathway [41]. The expression level of miR-206 was significantly downregulated in analyzed MPM tumor samples and cell lines compared to normal pleural tissue. Upregulation of pathway agents (*VEGFA, EGFR, MET, IGF1R, KRAS, CCND1, CDK4,* and *CDK6*) was identified in MPM and relatively high *IGF1R, KRAS, CCND1,* or *CDK4* expression was found to be significantly associated with poor overall survival in the TCGA dataset. miR-206 re-expression led to significant inhibition of malignant features in vitro and in vivo, and suppression of KRAS signaling. 

Taken together, the published literature and our recent work indicate a thus-far underestimated role for KRAS pathway alterations in MPM. An explanation could be that a molecular subset of MPM is initiated by mutations (or alterations in general) of *KRAS* and/or its pathway components. These are then subsequently lost during cancer evolution due to genomic instability, or they might simply be missed during sampling and sequencing as a result of the applied procedures, as has been shown for lung tumors (Figure 3) [42]. An alternative scenario could be that KRAS pathway alterations develop late during MPM progression as sub-clonal events and are hence missed. Whatever their nature and timing during MPM evolution, *KRAS* alterations appear to be functionally important since they are able to single-handedly drive the murine mesothelium to MPM, even more profoundly so in combination with *TP53* alterations. The data support that KRAS pathway alterations are marked targets for therapy in a subset of patients with MPM, either as first-line therapy if they truly present tumor-initiating mutations, or as second-line therapy if they drive therapy-resistant *KRAS*-driven MPM subclones. Acquired therapy-resistance is a major concern nowadays, as it has been shown for *KRAS*-driven chemotherapy-resistant pancreatic cancers or *KRAS*^G12C^-inhibitors in several *KRAS*-driven cancer types, for example [43,44]. The proposed mechanism of chemoresistance described by Mukhopadhyay et al. is conferred by KRAS via alteration of cellular metabolism. This role of KRAS as a driver of metabolic changes, such as the Warburg effect for instance, has been frequently reported for several cancers [45,46]. A deeper and extended understanding of the role of KRAS pathway alterations in MPM could open the door to new advances and insights that have not been elucidated before. This bears the opportunity of a new perspective, potential targets, and therapeutic possibilities.

Collectively, the data call for the deployment of more sensitive methods for the detection of KRAS pathway alterations in MPM, such as digital droplet PCR and maximal depth sequencing [18,47]. In addition, the future clinical identification of such low allelic frequency changes will hopefully facilitate the design of prospective clinical trials of KRAS signaling blockades against a significant fraction of human MPM that is driven by KRAS pathway alterations.

## 5. Conclusions 

The published literature, our recent work, and the analysis of publicly available data presented in this review indicate a thus-far underestimated role for KRAS pathway alterations in MPM. KRAS pathway changes and signaling warrant further investigations in the future, as they appear to be important in a significant proportion of MPM patients.

## Figures and Tables

**Figure 1 cancers-14-04303-f001:**
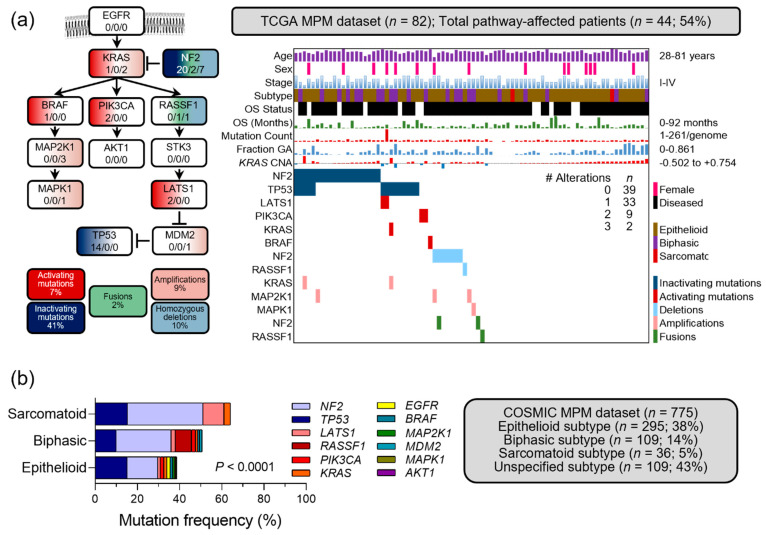
*KRAS* pathway alterations in The Cancer Genome Atlas (TCGA) and the Catalogue of Somatic Mutations in Cancer (COSMIC) MPM datasets. (**a**) A biological *KRAS* pathway as proposed by Matallanas et al. in 2011 [22]. Shown are its 13 genes (boxes) interconnected via activating (arrows) and inhibitory (dead-end) signaling events, together with color-coded alteration types and frequencies (legend). Numbers in boxes denote the numbers of TCGA MPM patients (*n* = 82 with full data) with mutations/fusions/copy number alterations for each gene. (**b**) Clinical and molecular data summary of TCGA MPM patients with KRAS pathway genes, color-coded clinical and molecular data plot (heatmap), number of patients with no, one, two, or three pathway alterations (table insert), and legend. OS, overall survival; GA, genome altered; CNA, copy number alteration. Raw data shown as patient numbers (*n*) and percentages (%) from Hmeljak et al., 2018 [4], were retrieved from https://www.cbioportal.org/ (accessed on 3 March 2022) using permanent link https://bit.ly/3BypsnC (accessed on 3 March 2022), and were manually analyzed and visualized on Microsoft Excel and PowerPoint. KRAS pathway mutation frequencies in MPM from COSMIC, stratified by histologic subtype (available at https://cancer.sanger.ac.uk/cosmic/browse/tissue?wgs=off&sn=pleura&ss=all&hn=mesothelioma&sh=&in=t&src=tissue&all_data=n (accessed on 3 March 2022); *n* = 775 patients). Shown are data summary and table, presented as mutation numbers (*n*) and frequencies (%). Note the gradually increasing cumulative mutation frequency of the pathway in biphasic and sarcomatoid MPM compared with epithelioid MPM. *p*, probability, 2-way ANOVA.

**Figure 2 cancers-14-04303-f002:**
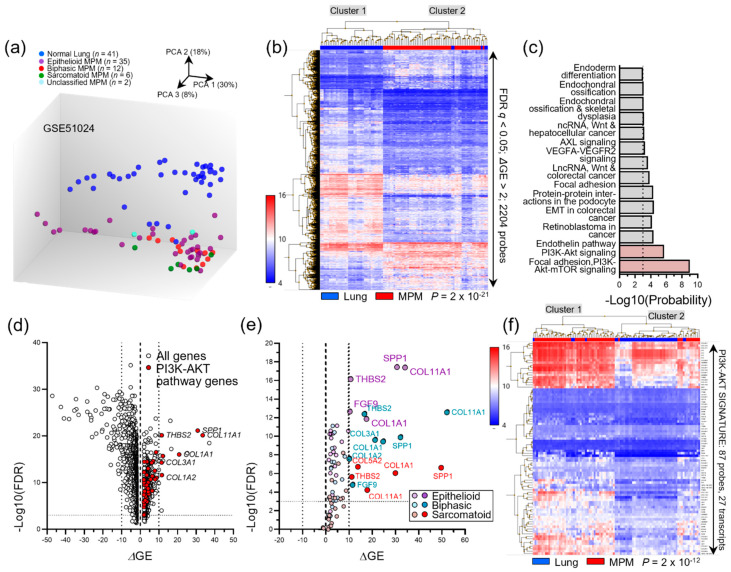
Transcriptional activation of the phosphoinositide 3-kinase (PI3K)-protein kinase (AKT) pathway downstream of KRAS in the Gene Expression Omnibus (GEO) MPM dataset GSE51024. Raw data of the gene expression profiles of MPM (*n* = 55) and normal lung (*n* = 41) tissues assessed by Affymetrix HG-U133_Plus_2 microarrays were retrieved from https://www.ncbi.nlm.nih.gov/geo/query/acc.cgi?acc=GSE51024 (accessed on 3 March 2022) and were analyzed using Affymetrix transcriptome analysis console v4.0. (**a**) Principal component analysis (PCA) plot showing color-coded individual patients (circles), percentile weight of the three principal components (%), patient numbers (*n*), and color-coded legend. (**b**) Unsupervised hierarchical clustering of all samples (columns) by 2204 probes (rows) differentially regulated in MPM and normal lung tissues. Data are presented as color-coded heatmaps of log2(signal intensities) produced via robust multi-array average normalization. FDR *q*, probability, false discovery rate; ΔGE, differential gene expression, fold-change MPM over lung tissues; *p*, probability, hypergeometric test. (**c**) Fourteen WikiPathways statistically significantly (*p* < 0.001, FDR) differentially regulated in the 2204 probes from (**b**). Data are presented as average WikiPathway significance (bars) and threshold (*p* < 0.001; dotted line). Red bars denote PI3K-AKT pathways and grey bars all other pathways. (**d**) Volcano plot of probes for all genes (white circles) and of 48 probes for 27 genes of WikiPathways “phosphoinositide 3-kinase (PI3K)-protein kinase (AKT) signaling” and “focal adhesion-PI3K-AKT-mTOR signaling” (red circles; including *CCNE2*, *CDK6*, *COL11A1*, *COL1A1*, *COL1A2*, *COL3A1*, *COL5A1*, *COL6A1*, *COL6A2*, *COL6A3*, *COMP*, *EFNA5*, *EFNA5*, *FGF18*, *FGF9*, *FN1*, *HSP90B1*, *IGF1*, *IGF2*, *ITGB4*, *LAMA1*, *PDGFD*, *SPP1*, *THBS2*, *THBS3*, *THBS4*, and *VTN*; hereafter called PI3K-AKT signature). (**e**) Volcano plot of differential gene expression of PI3K-AKT signature genes in the major histologic MPM subtypes. In (**d**,**e**), data are presented as color-coded individual probe data points (circles), thresholds of significance (dotted lines), and ΔGE = 0 reference (dashed lines). In (**e**), light colors denote non-significant and dark colors significant probes. (**f**) Unsupervised hierarchical clustering of all samples (columns) by 48 probes for 27 PI3K-AKT signature genes (rows). Data are presented as color-coded heatmaps of log2(signal intensities) produced via robust multi-array average normalization. FDR *q*, probability, false discovery rate; ΔGE, differential gene expression, fold-change MPM over lung tissues; *p*, probability, hypergeometric test.

**Figure 3 cancers-14-04303-f003:**
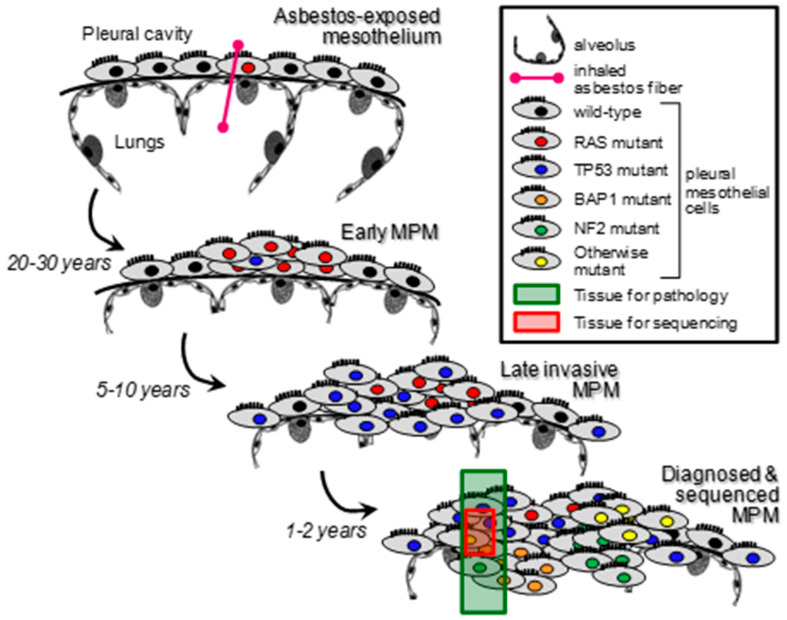
Schematic representation of the scenario for KRAS pathway alterations missed by next generation sequencing studies via sampling and allelic frequency bias. The sporadic nature of KRAS pathway alterations in MPM is compatible with both their possible early tumorigenicity, as well as with late clonal or sub-clonal natures. Given their low allelic frequency, however, the most likely explanation of the findings presented here is the later scenario, coupled with sampling bias.

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
