# Peer review of "KRAS Pathway Alterations in Malignant Pleural Mesothelioma: An Underestimated Player"

_cancers, 2022, doi:10.3390/cancers14174303_

Round 1
Reviewer 1 Report
Dr. Trassl and colleagues present their analysis of KRAS in mesothelioma. This is an interesting topic and a thoughtful assessment of the existing literature. Overall, the manuscript is clear but would benefit from some grammatical editing.
I have the following specific comments:
1) Despite the restriction of asbestos use in many countries, the historical presence of asbestos is still a problem. For example, while asbestos isn’t used in new homes in the US, renovations of older homes is a major ongoing exposure source. Please update the introduction to reflect this exposure source.
2) It is unclear to me what the review of the tumorigenesis of mesothelioma by asbestos adds to this manuscript.
3) The introduction is overly negative and paints a very grim, dismal and inaccurate perspective of mesothelioma. Please consider including a clinical with expertise treating this heterogeneous disease.
4) Discussion of the Checkmate 743 regimen of ipi/nivo that has become a front-line standard is missing.
5) There is an overemphasis of surgical paradigms in the introduction which are applicable to only a subgroup of those diagnosed with mesothelioma.
6) Throughout the body of this manuscript it is difficult to identify what the authors are reporting of their own unpublished work versus including their previously reported work in this manuscript. Please clarify.
7) Overall the body of the manuscript could be shortened and more emphasis placed on summarizing prior work rather than recapitulating those results. Additionally, that summary would allow better alignment of the information included to support the final conclusions of the paper.
Reviewer 2 Report
A timely review article by Dr. Trassl and their group elaborates on the role of altered KRAS signaling in malignant pleural mesothelioma. This is a very well-written review article that gives an overview of the field and sheds light on therapeutics. A few things need to be addressed before it is ready for acceptance, they are as follows:
1. It has been shown how altered KRAS signaling affects cancer metabolism (PMID: 33870211 and PMID: 25878364). Authors should add a few lines on this aspect of metabolic alteration in KRAS-driven malignant pleural mesothelioma. This aspect will be a novel addition and probably future avenues to study further.
2. Authors should mention if any clinical trial is being run in KRAS-driven malignant pleural mesothelioma. That will give readers a translational aspect of this review.
3. Authors should add a few lines explaining "therapy-resistant KRAS driven MPM" since resistance is the major cause of concern nowadays. It will be worthwhile to discuss that topic a bit by adding a couple of lines in the discussion. Since KRAS has been shown to play a significant role in chemoresistance (PMID: 31911550 and PMID: 34161704)
Round 2
Reviewer 2 Report
All concerns have been addressed, ready for acceptance.